# Accurate Prediction of Anxiety Levels in Asian Countries Using a Fuzzy Expert System

**DOI:** 10.3390/healthcare11111594

**Published:** 2023-05-30

**Authors:** Mouz Ramzan, Muhammad Hamid, Amel Ali Alhussan, Hussah Nasser AlEisa, Hanaa A. Abdallah

**Affiliations:** 1Department of Computer Science, National College of Business Administration and Economics (NCBA&E), Lahore 54000, Pakistan; 2Department of Statistics and Computer Science, University of Veterinary and Animal Sciences, Lahore 54000, Pakistan; 3Department of Computer Sciences, College of Computer and Information Sciences, Princess Nourah bint Abdulrahman University, P.O. Box 84428, Riyadh 11671, Saudi Arabiahaleisa@pnu.edu.sa (H.N.A.); 4Department of Information Technology, College of Computer and Information Sciences, Princess Nourah bint Abdulrahman University, Riyadh 84428, Saudi Arabia

**Keywords:** anxiety, anxiety prediction, fuzzy logic, fuzzy inference system

## Abstract

Anxiety is a common mental health issue that affects a significant portion of the global population and can lead to severe physical and psychological consequences. The proposed system aims to provide an objective and reliable method for the early detection of anxiety levels by using patients’ physical symptoms as input variables. This paper introduces an expert system utilizing a fuzzy inference system (FIS) to predict anxiety levels. The system is designed to address anxiety’s complex and uncertain nature by utilizing a comprehensive set of input variables and fuzzy logic techniques. It is based on a set of rules that represent medical knowledge of anxiety disorders, making it a valuable tool for clinicians in diagnosing and treating these disorders. The system was tested on real datasets, demonstrating high accuracy in the prediction of anxiety levels. The FIS-based expert system offers a powerful approach to cope with imprecision and uncertainty and can potentially assist in addressing the lack of effective remedies for anxiety disorders. The research primarily focused on Asian countries, such as Pakistan, and the system achieved an accuracy of 87%, which is noteworthy.

## 1. Introduction

Anxiety, sociopathy, emotional stability, and desire are just a few psychological factors affecting human behavior [1]. Human behavior can also be described in terms of different patterns, including the thoughts and emotions through which each person adapts to the circumstances of their existence. Another factor that affects behavior is anxiety, described as an unpleasant unease of the mind related to an imminent or expected illness. It symbolizes an internal rather than an exterior threat or danger [2]. Anxiety is a disorder that can be brought on by ongoing stress and if it is, sufferers may face significant hazards. While some disorders only manifest in adulthood, anxiety can start as early as childhood and threaten the individual and the larger community. According to reports [3,4], anxiety can negatively affect up to one-third of the population. In light of this observation, the WHO estimates that 450 million individuals globally experience stress and anxiety [5]. Moreover, the characteristics of emotional integrity that are highly complex and associated with an individual’s well-being are directly linked to their quality of life. Stress is a frequently cited factor, and extensive research has been conducted on its diagnosis and treatment [6]. However, the diagnosis of an anxiety disorder is a challenging and intricate process [7]. Anxiety is highly neglected in Asian countries, such as Pakistan [8,9], and it is often not taken seriously. However, this can lead to serious health hazards because diseases such as heart attacks and brain hemorrhages are related to anxiety. People must take anxiety symptoms seriously and observe their surroundings to identify whether someone has anxiety symptoms. The early treatment of anxiety can save a person’s life.

Various techniques can be used to predict anxiety at early stages [10]. One of these techniques is the development of a fuzzy-rule-based system, which can predict the disorder based on the created knowledge base. The development of a machine-learning model is another approach that can be utilized to predict anxiety. Different machine-learning techniques are available, such as SVM, decision trees, and ANN. However, problems can arise due to a lack of inputs or low accuracy in the system.

The importance of the fuzzy expert system lies in its ability to predict anxiety at an early stage. It is a useful tool for healthcare professionals to identify the risk of anxiety disorder and to provide early intervention to patients. The system utilizes a knowledge base, which makes it more reliable and accurate in predicting anxiety. On the other hand, the use of machine-learning techniques also provides a promising approach to the prediction of anxiety, but the accuracy of the model is dependent on the quality and quantity of the input data. The questions that arose after learning more about the issue were:

**RQ1.** Which symptoms can lead to anxiety in a patient?

**RQ2.** How much accuracy can be obtained by predicting anxiety at certain levels?

To answer these research questions, our research objectives and key research contributions are outlined below.

A fuzzy inference system (FIS) is proposed, taking the symptoms as inputs using them to predict outcomes. Patients suffering from anxiety report their basic symptoms to the system. The system uses these symptoms as inputs and predicts whether the patients are suffering from anxiety. The study offers a workable method for the precise diagnosis and prediction of anxiety while advancing our understanding of anxiety and its effects on people and society. The early diagnosis and treatment of anxiety can have major advantages because they can improve treatment outcomes and lessen the toll the condition takes on sufferers, their families, and society. In its early stages, when symptoms may be subtle or ambiguous, anxiety is a complicated disorder that can present in various ways and may not always be obvious. As a result, the creation of a fuzzy expert system that can reliably predict anxiety at an early stage may be extremely helpful in assisting medical practitioners in making prompt and knowledgeable therapeutic decisions.

The rest of this paper is organized as follows. In Section 2, a review of the literature is discussed. Next, the detailed methodology is described, in Section 3. In Section 4, demonstrations and evaluations are presented, while Section 5 presents the answers to the research question. Finally, in Section 6, concludes the research and provides directions for future work.

## 2. Literature Review

In recent years, systems based on artificial intelligence (AI) have been increasingly utilized to improve the quality, sensitivity, and timeliness of the diagnosis of psychiatric disorders, including anxiety disorders. These systems employ AI techniques such as fuzzy logic, neural networks (NNs), support vector machines (SVMs), and decision trees (DTs) [11,12,13]. Altintaş et al. [14] conducted a review of research on machine-learning techniques for diagnosing anxiety disorders between 2015 and 2021. The study examined databases for information on different categories of anxiety disorder and identified thirty different ML techniques used in their research. They compared these techniques in terms of sample size, age, chosen techniques, best practices, and performance values. The authors found that the random forest algorithm (RFA) was the most commonly used ML technique, ensuring the best results and accuracy. The authors also suggested that AI techniques can provide valuable information to researchers and clinicians in areas such as personalized treatment, diagnosis, and prognosis, given the heterogeneity of the data obtained from anxiety patients. Furthermore, Lotfi et al. [15] proposed a system for monitoring and managing anxiety among young people using machine learning. The authors describe the development of a mobile application that collects data on users’ anxiety levels and uses machine-learning algorithms to predict and manage anxiety. Their system aims to provide an effective and efficient approach to identifying and managing anxiety among young people. Their research concludes that the proposed system improves the quality of life of young people and facilitates the early detection and management of anxiety disorders. Furthermore, Kumar et al. [16], developed a new supervised learning-based prediction model, an Anxious Depression (AD) prediction model, for efficiently predicting anxious depression in real-time tweets. Based on the user’s posting patterns and linguistic cues, the feature set is defined using a five-tuple vector <w, t, f, s, c>. The representation of each entry is introduced as follows:

< w: word >: The presence or absence of the anxiety-related word.

< t: timing >: More than two posts during odd night hours.

< f: frequency >: More than thirty posts in an hour.

< s: sentiment >: On average, more than 25% of posts over 30 days with negative polarity.

< c: contrast >: The presence of a polarity contrast of more than 25% in posts within the past twenty-four hours.

Moreover, Susanto et al. [17] proposed a fuzzy-logic-based model to predict mathematics anxiety in students. The model was constructed using a dataset of 470 Turkish university students who completed a mathematics-anxiety scale and a demographic questionnaire. The study examined the relationship between mathematics anxiety and various demographic factors, such as gender, age, and major. The model was constructed using a Mamdani-type fuzzy inference system, and its performance was evaluated using various statistical measures. The results showed that the fuzzy model can accurately predict mathematics anxiety in students and can be useful for educators and researchers to identify students at risk of developing mathematics anxiety.

Further, Khullar et al. [18] proposed a method for detecting anxiety based on physiological signals. Their study used an electrocardiogram (ECG) and galvanic skin response (GSR) signals to develop a machine-learning model for anxiety detection. The proposed model uses an ensemble approach that combines multiple machine-learning algorithms, such as decision trees, random forest, and support vector machine. The performance of the proposed model was evaluated using a dataset of 18 subjects, who were exposed to anxiety-inducing stimuli. The results showed that the model accurately detects anxiety based on physiological signals. The authors suggested that the proposed approach can be further explored to better predict anxiety levels.

Furthermore, Devi et al. [19] proposed a method for predicting the anxiety levels of students using an adaptive neuro-fuzzy-inference system (ANFIS). The study used a dataset of 208 undergraduate students who completed the State-Trait Anxiety Inventory (STAI) questionnaire. The proposed ANFIS model considers various demographic and academic factors, such as age, gender, major, academic performance, and STAI score. The performance of the proposed model was evaluated using various statistical measures, such as mean absolute error and root-mean-squared error. The results showed that the proposed ANFIS model achieved high accuracy in predicting the anxiety levels of the students. The authors further suggested that a developed system should be further explored and improved for the better prediction of anxiety.

From the literature review presented above, we can conclude that anxiety symptoms can be different from person to person and that they can also be different across regions. For example, people in Asian countries, such as Pakistan, may show different anxiety symptoms. Therefore, it is necessary to create a fuzzy logic system that can target the Asian continent in particular, in order to predict the disorder correctly. To this end, we will used symptoms widely shown in patients with anxiety in Pakistan to build our fuzzy expert system. Figure 1 represents the core symptoms of anxiety.

## 3. Methodology

### 3.1. Fuzzy Inference System

A FIS system uses fuzzy logic to map input and output variables based on rules. An expert system can make decisions or predictions based on fuzzy reasoning. The FIS consists of three main components: the fuzzifier, the inference engine, and the defuzzifier. The fuzzifier converts input data, which may be crisp (i.e., exact) or fuzzy (i.e., imprecise), into fuzzy sets representing the degree of membership in a particular category. The inference engine applies a set of rules defined by experts or generated automatically from data to determine the appropriate output based on the input data. Finally, the defuzzifier converts the fuzzy output into a crisp output value. The FIS system can be used in various applications, such as control, decision-making, and pattern-recognition systems. They are particularly useful in studies with high levels of uncertainty or imprecision in their data.

The proposed system uses fuzzy logic as the ideal methodology for decision assistance. Fuzzy logic was chosen because it enables flexible and nuanced decision making through the IF-ELSE-based methodology. It can handle values in points, which conventional rule-based systems cannot. In this study, which focused on predicting anxiety levels, fuzzy logic’s capacity to collect and interpret data more granularly was beneficial and in line with the study’s goals. Therefore, fuzzy logic was chosen for its usefulness, its compatibility with the study’s goals, and its ability to handle values in points.

### 3.2. Fuzzy Inference System for Prediction of Anxiety Levels

Our proposed system can predict anxiety at different levels (no, low, mild, and high). The fuzzy system’s architecture is depicted in Figure 2. The fuzzification process creates fuzzy logic sets from crisp inputs. Based on the input given, the inference engine analyzes the matched set.

The input and output of the fuzzy system are depicted in Figure 3. The fuzzy sets that defuzzification uses as the inference engine transform inputs to produce the results. Figure 4 visually represents the system flowchart, highlighting the sequence of steps and decision points involved in its operation. Figure 4 serves as a useful tool for understanding the system’s functionality and identifying potential areas for improvement.

### 3.3. Knowledge-Base Input-and-Output Fuzzy System

Five symptoms were taken as input variables and one output variable was used for this system simulation. Fuzzy inputs Nervousness and Panic had three fuzzy terms (low, mild and high), while inputs sweating, trembling, and increased heart rate had two fuzzy terms (no and yes). Creation of FIS rules was executed with the guidance of experts in the field. The rules were defined using a combination of linguistic terms and mathematical expressions. In order to accurately define the rules, a thorough analysis of the available linguistic terms was conducted, and they were mathematically combined to form a comprehensive set of rules. The linguistic terms were categorized based on the number of values they represented to facilitate the creation of these rules. This was achieved by separating the terms with two values from those with three. In this way, we effectively managed the vast array of terms and created precise rules. The rules were defined using a mathematical expression resulting from multiplying the number of linguistic terms with two values by the number of terms with three. This expression yielded a total of 72 rules, each of which was carefully crafted to ensure accuracy and reliability in the FIS system. Overall, creation of these rules was a rigorous and meticulous process involving the input of domain experts and the use of advanced mathematical techniques.

Further details are shown in Table 1. The parameter regarding the fuzzy set describes the range of levels of symptoms. As we already know, the range is from 0–1, which means that when the symptom has three values, the range is divided into three equal sizes. If the symptom has two values, the range is divided into two. Furthermore, the symptom level is shown based on the range value provided.

### 3.4. Membership Functions

Membership functions are essential in fuzzy logic as they represent an element’s degree of membership in a fuzzy set. This is important because fuzzy sets have continuous boundaries, which differ from crisp sets with clear boundaries. Membership functions enable the representation of uncertainty and imprecision in the input data, facilitating more flexible and nuanced decision-making. They are commonly used to perform mathematical operations on a given dataset and are applicable across different platforms. Table 2 describes the membership functions employed in the proposed system.

### 3.5. Defuzzification

Defuzzification is the process of converting fuzzy logic sets and related membership degrees into a calculable or quantifiable output in crisp logic. Fuzzy logic is a form of reasoning that deals with imprecise or uncertain information, while crisp logic deals with precise or certain information. Defuzzification takes the fuzzy input and output values from a fuzzy set and converts them into a crisp value that can be used for decision making. There are several methods for defuzzification, including the centroid method, mean-of-the-maximum method, and weighted-average method. Each method has advantages and disadvantages, and the choice of method depends on the specific application and the desired level of accuracy.

Figure 5, Figure 6 and Figure 7 show graphical representations of the defuzzification. Figure 5 shows the control surface; the yellowish color represents the anxiety level, bluish represents the sweating, and greenish represents the nervousness in the patient. We can see that anxiety levels gradually increased after giving two predictors, sweating and nervousness. Figure 6 shows the control surface, in which the yellowish color represents the anxiety levels, bluish represents the panic, and light-blue color represents increased heart rate in any patient. We can see that after giving another two predictors, panic and increased heart rate, the anxiety level reached a new value. Figure 7 shows the control surface, in which the yellowish color represents the anxiety levels, bluish color represents the trembling, and the greenish color represents panic in any patient. Here, we can see that the anxiety level drastically increased after giving symptoms of trembling and panic. This illustrates how different symptoms can be mapped to different fuzzy sets and how defuzzification can convert these fuzzy sets into crisp values for decision making.

## 4. Demonstration and Evaluation

The demonstration-and-evaluation phase consisted of three activities. The experiment results generated using MATLAB were discussed in the first activity, while the second activity featured the prediction of anxiety by the fuzzy expert system. The third activity featured a detailed description of the comparative evaluation of the proposed system and a discussion.

### 4.1. Experimental Results

MATLAB is a widely used tool in scientific computing, image processing, data visualization, and numerical computation. In this study, it was used to calculate the experimental results. The figures, specifically Figure 8, Figure 9, Figure 10 and Figure 11, illustrate the use of MATLAB in the determination of anxiety levels based on certain factors, such as nervousness, sweating, trembling, increased heart rate, and panic.

Figure 8 outlines the criteria for assessing anxiety levels without any or with low factors. If nervousness, sweating, trembling, increased heart rate, and panic were absent or low, the anxiety level was considered as “No.”

Figure 9 outlines the criteria for assessing anxiety levels when some factors are present. If nervousness is low, sweating is absent, trembling is present, increased heart rate is not present, and panic is mild, then the anxiety level is considered “Low.”

Figure 10 outlines the criteria for assessing anxiety levels when several factors are present. If nervousness is mild, sweating, trembling, and increased heart rate are present, and panic is mild, then the anxiety level is considered “Mild.”

Figure 11 outlines the criteria for assessing anxiety levels when most factors are high. Specifically, if nervousness, sweating, trembling, increased heart rate, and panic are all high, then the anxiety level is considered “High.” In summary, MATLAB was utilized to calculate the experimental results. Figure 8, Figure 9, Figure 10 and Figure 11 demonstrate the use of MATLAB in the determination of the anxiety levels based on specific factors, such as nervousness, sweating, trembling, increased heart rate, and panic. We set the range from 0 to 1. If we changed the value of ‘No’ or any other symptom to “Mild” or “High,” the anxiety levels changed accordingly. This is because the levels were solely based on the input ranges. The figures below demonstrate how the different range values affected the anxiety levels.

### 4.2. Anxiety-Prediction Fuzzy Expert System

The anxiety-prediction system uses various anxiety-related symptoms as inputs from the user. These symptoms include nervousness, sweating, trembling, increased heart rate, and panic. By analyzing these inputs, the system can predict the user’s anxiety levels, which can be categorized into four levels: no anxiety, low anxiety, mild anxiety, and high anxiety. The proposed system can be visualized as shown in Figure 12, which demonstrates how the inputs from the user are processed and used to determine the user’s anxiety levels. The system is designed to provide a quick and accurate assessment of the user’s anxiety level, which can be used to inform treatment decisions or track progress over time.

### 4.3. Comparative Evaluation of the Proposed System

The data were collected from the research center of Shalimar Hospital in Lahore, Pakistan. The data were organized during the examination of patients in the outpatient department (OPD). Doctors diagnosed the disorder by collecting the symptoms, which were subsequently used to evaluate the accuracy and results of our system. Table 3 presents a comprehensive assessment of the performance of our proposed anxiety-prediction system by comparing the results obtained from the experts’ opinions, medical reports, and our system. To evaluate the accuracy of our system, we tested it on the patient data from Shalimar Hospital. We compared the results obtained from our system with the doctors’ reviews and the patients’ medical reports. By comparing the results from the expert opinions, medical reports, and our proposed system, we identified any discrepancies and assessed the overall accuracy of our system. The comparison also helped to identify areas in which the system may need further improvement. Our proposed system showed an accuracy of 87%. We determined the accuracy by comparing our results with the medical records, using the data for 15 patients. Our proposed system correctly predicted the anxiety levels of 13 patients. We estimated the accuracy using the formula (100/15 × 13). Table 3 provides a comprehensive assessment of the overall performance of our proposed system, which can be used to inform future developments and improve the system’s accuracy.

Patient 1 has high anxiety according to the expert opinion and medical reports; our system also accurately established that this patient has high anxiety. Furthermore, patient 2 has high anxiety according to the expert opinion and medical reports, and our system also accurately predicted the high anxiety of this patient. Patient 3 has no anxiety according to the expert opinion and medical reports, which our system also accurately predicted. Patient 4 has low anxiety according to the expert opinion and our proposed system, while the medical report suggests that the patient has mild anxiety. Patient 5 has no anxiety according to the expert opinion, the medical report, and our proposed system. The data were collected in a random manner. We evaluated a total of 15 patients in Table 3. A total of three patients showed high anxiety, five showed low anxiety, five showed no anxiety, and two showed mild anxiety.

We evaluated our system by comparing its results with expert opinions and the results from medical reports. The expert opinion represented the doctor’s diagnosis before any medical results were reported. The medical reports represented the results from the GAD test, which is essentially a test to examine anxiety disorder. After comparing the results of our system with expert opinions and medical reports, we determined the accuracy of our system. Table 3 compares three results from our system, the expert opinions, and the medical reports. It displays the anxiety levels predicted by each source for each patient. The proposed system exhibited an accuracy of 87%, accurately predicting the anxiety level of 13 out of 15 patients. We also compared these results with previous research on the same issue. This comparison with previous research indicated that the proposed system outperforms the models developed by Boukhechba et al. [21], Dhyani et al. [22], and Borse et al. [23] in terms of accuracy. The comparison of the results from different sources helped to identify discrepancies and assess the system’s overall accuracy. This study’s findings can be used to inform future developments and improve the accuracy of the system.

## 5. Answers to Research Questions

The answers to the research questions were formulated at the beginning of this research.

**RQ1.** Which symptoms can lead to anxiety in a patient?

Anxiety can manifest in many ways and the symptoms can vary from person to person. Some common anxiety symptoms include nervousness and panic, excessive sweating, trembling, increased heart rate, and restlessness.

In Asian countries, such as Pakistan, where there are high stress levels due to various factors, such as financial instability, political turmoil, and societal pressure, anxiety can be a prevalent issue. Additionally, cultural factors, such as the stigmatization of mental health issues and a lack of awareness and education, can further exacerbate the problem.

**RQ2.** How much accuracy can be obtained by predicting anxiety at certain levels?

Th prediction of anxiety at certain levels can be challenging due to the complex nature of the disorder and the variability of symptoms from person to person. Anxiety can occur in short-term and long-term forms and present differently depending on the individual. It is important to carefully select the symptoms to use as indicators of the disorder to predict anxiety at different levels accurately. A proposed system was developed in a research study to predict anxiety at four levels. The system used a combination of patient symptoms and medical reports to predict anxiety levels. The study’s results were then compared to patients’ medical reports and expert opinions. The system was found to have an accuracy of 87% in predicting anxiety levels. This suggests that the system accurately identified the different anxiety levels in the studied patients.

It should be noted that the system’s accuracy may vary depending on the population studied and the specific symptoms used as indicators of the disorder. Further research is needed to validate the system in different populations and to improve its accuracy. However, the results of this study provide a promising start in the development of effective tools for predicting anxiety at different levels.

## 6. Conclusions and Future Work

This paper proposed an expert system that utilizes fuzzy inference techniques to provide an objective and reliable method for detecting anxiety levels in patients early. The proposed system’s success in achieving an accuracy of 87% an Asian country, Pakistan, is noteworthy and provides hope for the early-stage diagnosis and treatment of anxiety disorders. The proposed system is an innovative approach that aims to predict the levels of anxiety in patients by using a combination of fuzzy enhanced rule-based sets and patient symptoms as inputs. Our results indicate that the system has a level of accuracy that is comparable to an expert opinion, which can potentially assist clinicians in diagnosing and treating anxiety disorders. However, a limitation of this work is that it only predicted anxiety and did not predict other psychological disorders, such as depression. In future work, the system’s scope can be expanded to predict the outputs based on the symptoms of both anxiety and depression, thereby providing a more comprehensive assessment of a patient’s mental health. By addressing the limitations of this work and incorporating more symptoms and relevant information, the system’s accuracy can be further improved, making it more useful to medical professionals in providing accurate assessments and treatment recommendations for patients. Overall, this expert system represents a promising tool for the early detection and treatment of anxiety disorders.

## Figures and Tables

**Figure 1 healthcare-11-01594-f001:**
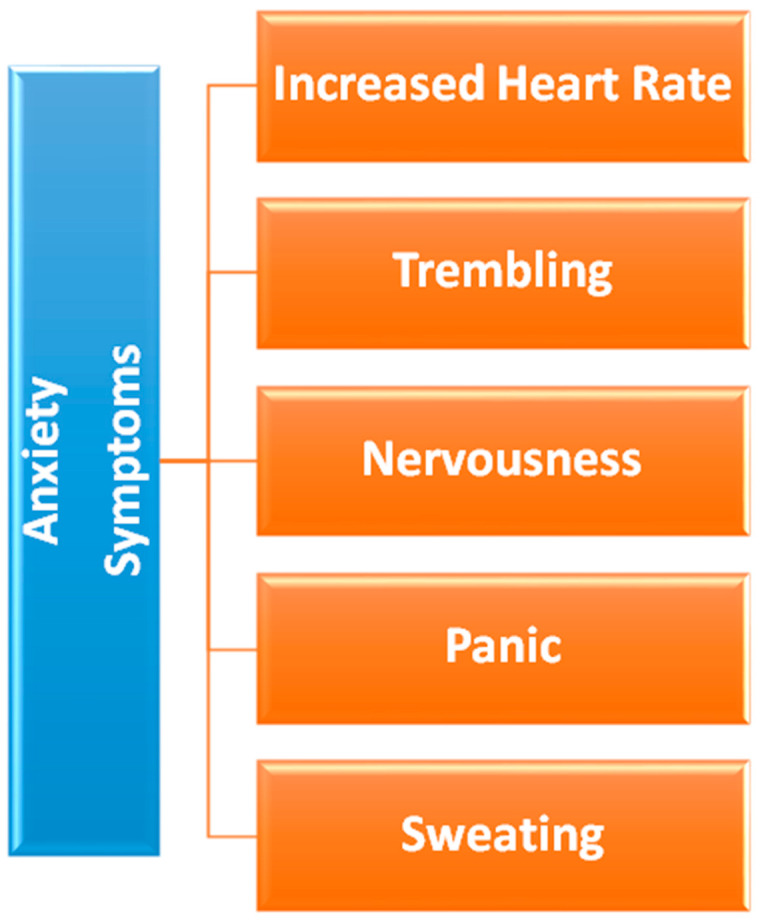
Anxiety symptoms [20].

**Figure 2 healthcare-11-01594-f002:**
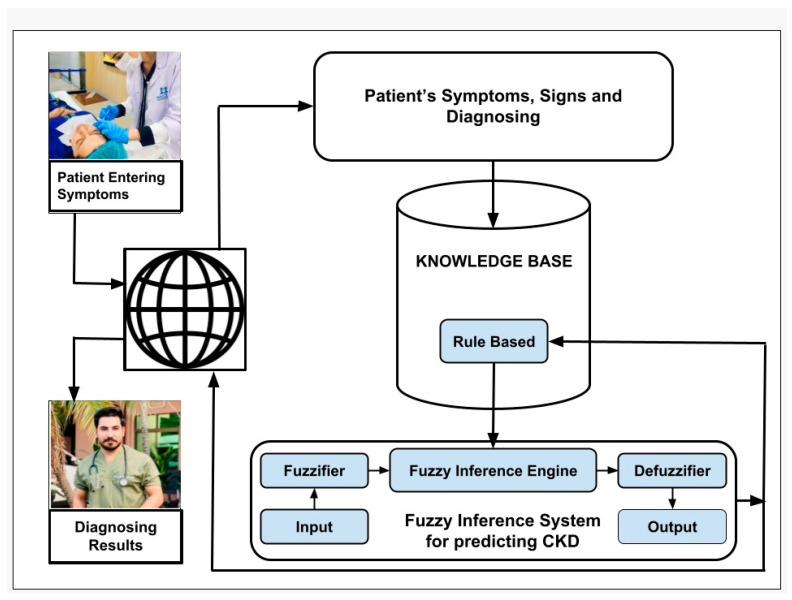
Fuzzy system architecture for predicting anxiety.

**Figure 3 healthcare-11-01594-f003:**
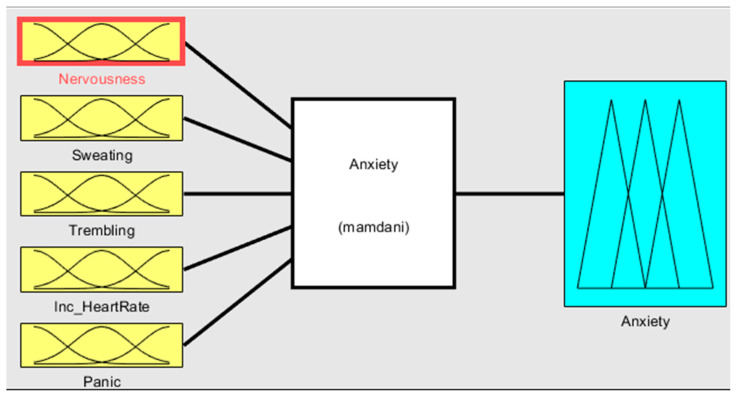
Input and output of the proposed system.

**Figure 4 healthcare-11-01594-f004:**
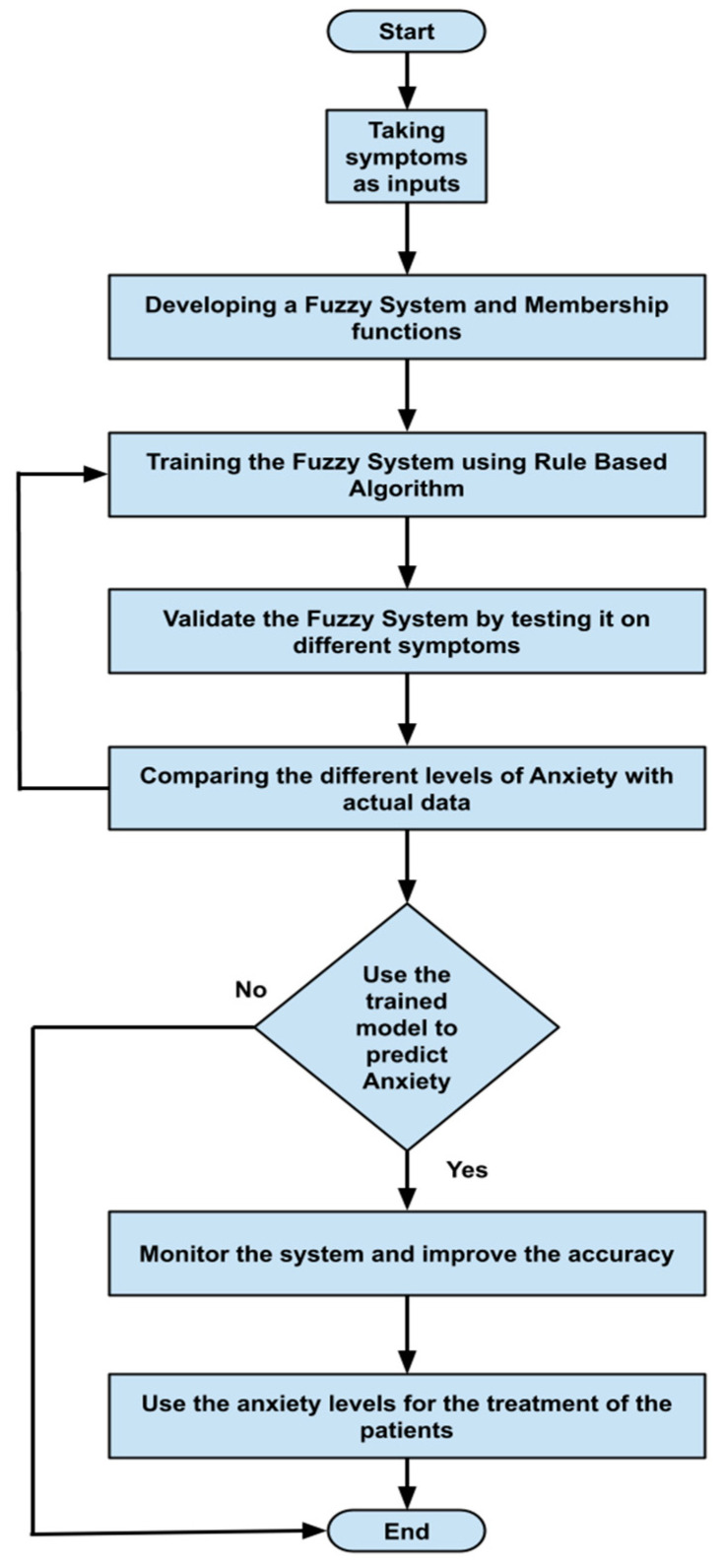
Flowchart of the proposed system.

**Figure 5 healthcare-11-01594-f005:**
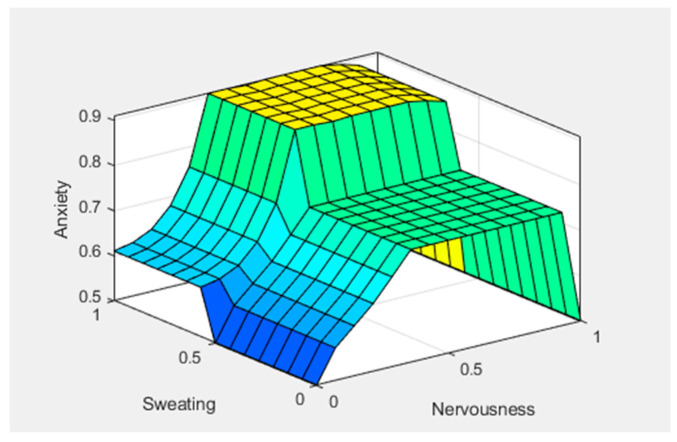
Surface inputs (X—Sweating and Y—Nervousness) and output (Anxiety).

**Figure 6 healthcare-11-01594-f006:**
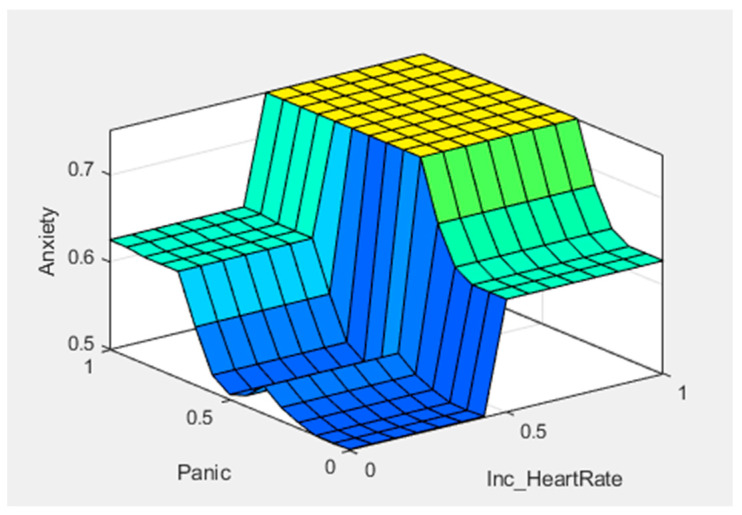
Surface inputs (X–Panic and Y—Increased Heart Rate) and output (Anxiety).

**Figure 7 healthcare-11-01594-f007:**
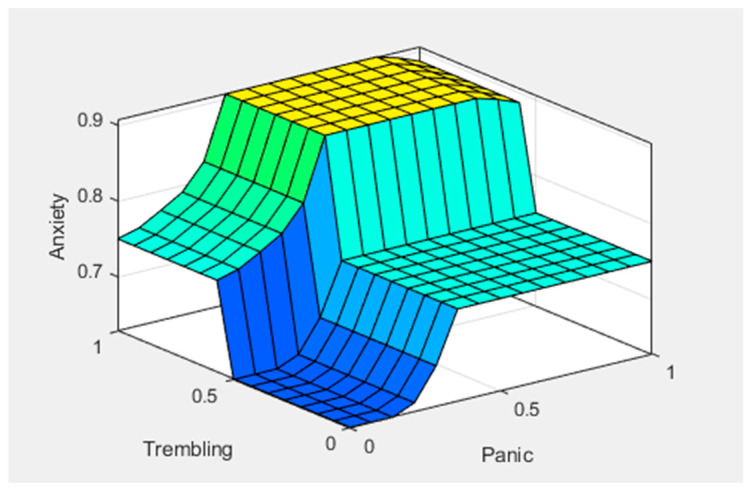
Surface inputs (X—trembling and Y—panic) and output (anxiety).

**Figure 8 healthcare-11-01594-f008:**
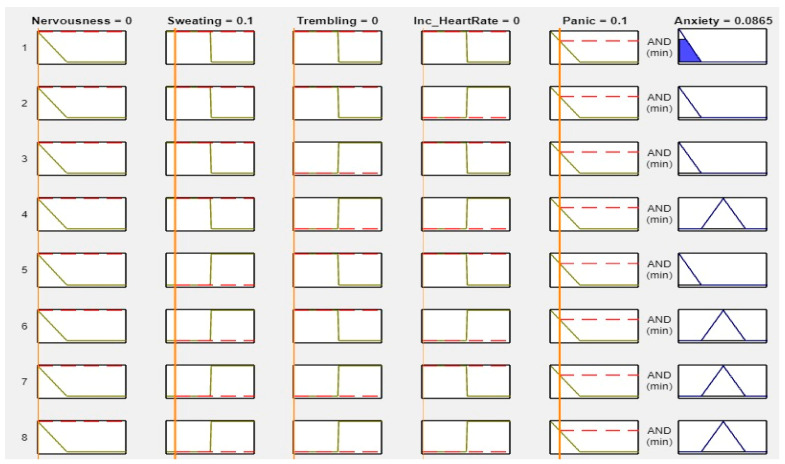
Lookup diagram for no anxiety.

**Figure 9 healthcare-11-01594-f009:**
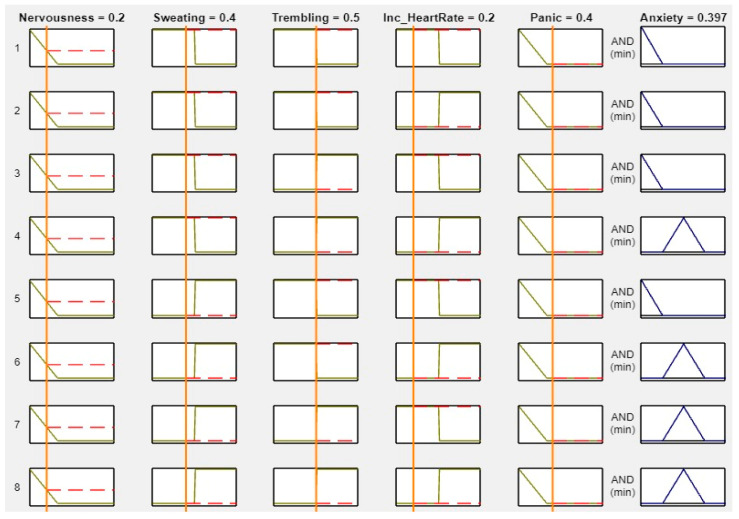
Lookup diagram for low anxiety.

**Figure 10 healthcare-11-01594-f010:**
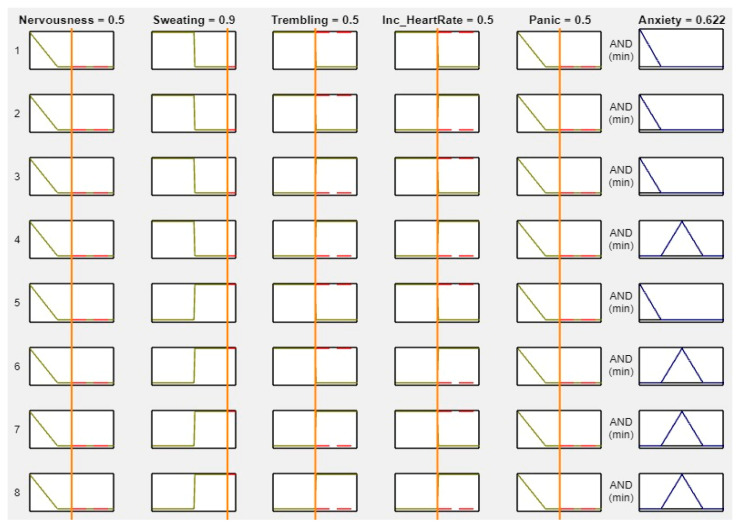
Lookup diagram for mild anxiety.

**Figure 11 healthcare-11-01594-f011:**
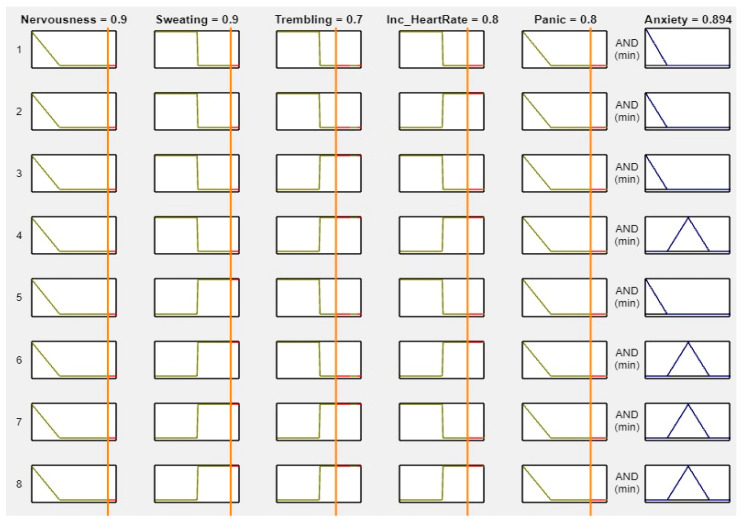
Lookup diagram for high anxiety.

**Figure 12 healthcare-11-01594-f012:**
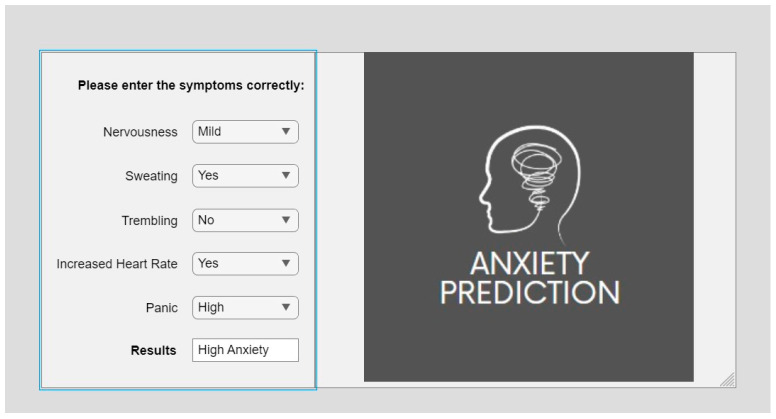
Proposed anxiety-prediction system.

**Table 1 healthcare-11-01594-t001:** Ranges of fuzzy systems for predicting anxiety.

Input and Output Variables	Linguistic Term	Parameters of Fuzzy Sets
Nervousness	{“Low”, “Mild”, “High”}	{[0, 0, 0.33], [0, 0.33, 0.66], [0.33, 0.66, 0.99]}
Sweating	{“No”, ”Yes”}	{[0, 0, 0.5, 0.51], [0.5, 0.51, 1, 1]}
Trembling	{“No”, ”Yes”}	{[0, 0, 0.5, 0.51], [0.5, 0.51, 1, 1]}
Increased Heart Rate	{“No”, ”Yes”}	{[0, 0, 0.5, 0.51], [0.5, 0.51, 1, 1]}
Panic	{“Low”, “Mild”, “High”}	{[0, 0, 0.33], [0, 0.33, 0.66], [0.33, 0.66, 0.99]}
Anxiety	{”No”, “Low”, “Mild”, “High”}	{[0, 0, 0.25], [0, 0.25, 0.50], [0.25, 0.50, 0.75], [0.50, 0.75, 1]}

**Table 2 healthcare-11-01594-t002:** Membership functions.

Input	Membership Functions	Graphical Representation of Membership Functions
Nervousness	$\mu_{low}\left(x\right)=\left\{\frac{0.33-x}{0.33}\;0\leq x\leq 0.33\right\}$ $\mu_{mild}\left(x\right)=\left\{\begin{array}{c} \frac{x}{0.33}\;0\leq x\leq 0.33\\ \frac{0.66-x}{0.33}\;0.33\leq x\leq 0.66 \end{array}\right\}$ $\mu_{high}\left(x\right)=\left\{\begin{array}{c} \frac{x}{0.66}\;0\leq x\leq 0.66\\ \frac{0.99-x}{0.33}\;0.66\leq x\leq 0.99 \end{array}\right\}$	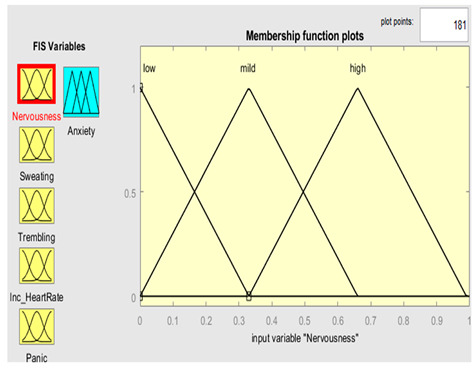
Sweating	$\mu_{no}\left(y\right)=\left\{\begin{array}{c} \frac{y-0}{0}\;0\leq y\leq 0.51\\ \frac{0.52-y}{0.01}\;0.51\leq y\leq 0.52 \end{array}\right\}$ $\mu_{yes}\left(y\right)=\left\{\begin{array}{c} y-0.51\;0\leq y\leq 0.51\\ \frac{1-y}{0}\;0.51\leq y\leq 0.52 \end{array}\right\}$	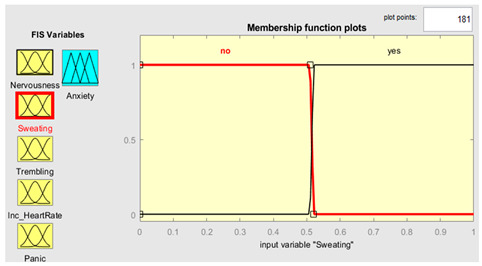
Trembling	$\mu_{no}\left(t\right)=\left\{\begin{array}{c} \frac{t-0}{0}\;0\leq t\leq 0.5\\ \frac{0.51-t}{0.01}\;0.5\leq t\leq 0.51 \end{array}\right\}$ $\mu_{yes}\left(t\right)=\left\{\begin{array}{c} t-0.5\;0\leq t\leq 0.5\\ \frac{1-t}{0}\;0.5\leq t\leq 0.51 \end{array}\right\}$	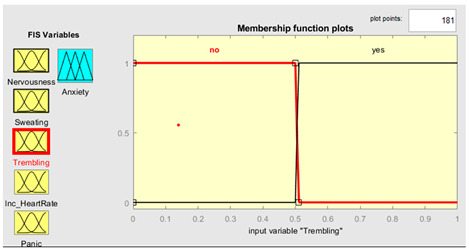
Increased Heart Rate	$\mu_{no}\left(u\right)=\left\{\begin{array}{c} \frac{u-0}{0}\;0\leq u\leq 0.49\\ \frac{0.5-u}{0.01}\;0.49\leq u\leq 0.50 \end{array}\right\}$ $\mu_{yes}\left(u\right)=\left\{\begin{array}{c} u-0.49\;0\leq u\leq 0.49\\ \frac{1-u}{0}\;0.49\leq u\leq 0.5 \end{array}\right\}$	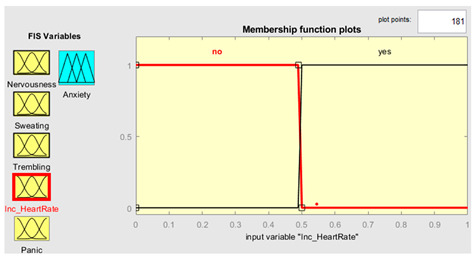
Panic	$\mu_{low}\left(x\right)=\left\{\frac{0.35-x}{0.35}\;0\leq x\leq 0.35\right\}$ $\mu_{mild}\left(x\right)=\left\{\begin{array}{c} \frac{x}{0.35}\;0\leq x\leq 0.35\\ \frac{0.7-x}{0.35}\;0.35\leq x\leq 0.7 \end{array}\right\}$ $\mu_{high}\left(x\right)=\left\{\begin{array}{c} \frac{x}{0.7}\;0\leq x\leq 0.7\\ \frac{1.05-x}{0.35}\;0.7\leq x\leq 1.05 \end{array}\right\}$	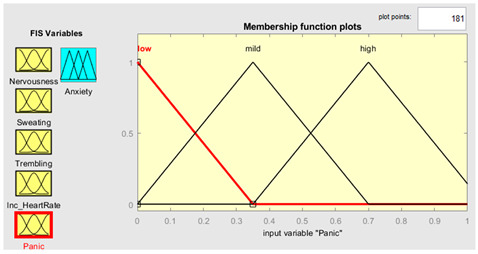
Anxiety	$\mu_{no}\left(z\right)=\left\{\frac{0.25-z}{0.25}\;0\leq z\leq 0.25\right\}$ $\mu_{low}\left(z\right)=\left\{\begin{array}{c} \frac{z}{0.25}\;0\leq z\leq 0.25\\ \frac{0.5-z}{0.25}\;0.25\leq z\leq 0.5 \end{array}\right\}$ $\mu_{mild}\left(z\right)=\left\{\begin{array}{c} \frac{z}{0.5}\;0\leq z\leq 0.5\\ \frac{0.75-z}{0.25}\;0.5\leq z\leq 0.75 \end{array}\right\}$ $\mu_{high}\left(z\right)=\left\{\begin{array}{c} \frac{z}{1}\;0\leq z\leq 1\\ \frac{1-z}{0.25}\;0.75\leq z\leq 1 \end{array}\right\}$	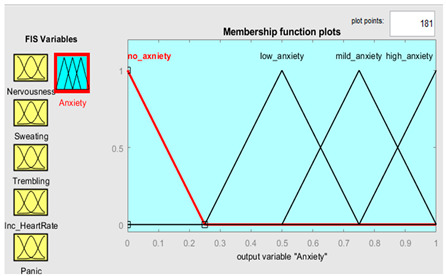

**Table 3 healthcare-11-01594-t003:** Comparative evaluation.

Patients	Expert Opinions	Medical Reports	Proposed System
Patient 1	High Anxiety	High Anxiety	High Anxiety
Patient 2	High Anxiety	High Anxiety	High Anxiety
Patient 3	No Anxiety	No Anxiety	No Anxiety
Patient 4	Low Anxiety	Mild Anxiety	Low Anxiety
Patient 5	No Anxiety	No Anxiety	No Anxiety
Patient 6	Low Anxiety	Low Anxiety	Low Anxiety
Patient 7	Low Anxiety	Low Anxiety	Low Anxiety
Patient 8	Low Anxiety	Low Anxiety	Low Anxiety
Patient 9	No Anxiety	No Anxiety	No Anxiety
Patient 10	Mild Anxiety	Mild Anxiety	Mild Anxiety
Patient 11	No Anxiety	No Anxiety	No Anxiety
Patient 12	High Anxiety	High Anxiety	High Anxiety
Patient 13	High Anxiety	High Anxiety	Mild Anxiety
Patient 14	Low Anxiety	Low Anxiety	Low Anxiety
Patient 15	No Anxiety	No Anxiety	No Anxiety

## Data Availability

Not applicable.

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
