# Peer review of "Accurate Prediction of Anxiety Levels in Asian Countries Using a Fuzzy Expert System"

_healthcare, 2023, doi:10.3390/healthcare11111594_

Round 1
Reviewer 1 Report
The motivation of the study shall be highlighted. Moreover, the method used shall be illustrated in detail, with a comparison to other approaches.
Author Response
Reviewer#1, Concern # 1:
The motivation of the study shall be highlighted. Moreover, compared to other approaches, the method used shall be illustrated in detail.
Author response:
Thank you for reading our paper so carefully, we really appreciate it. The motivation behind our research and using a fuzzy decision support system has been illustrated in detail.
Author action: We updated the manuscript by Page 2, Section 1.
Reviewer 2 Report
Dear Authors,
The content of the article is hot in recent research and its application field is interesting. The article is well designed overall. However, I think there are points that need to be corrected:
1. The fuzzy numbers in column 3 of Table 1 are difficult to understand. It would be easier to read if commas were used.
2. If brief information about fuzzy logic was given, the article would be easier to follow, especially for readers unfamiliar with fuzzy systems. What is the membership function, what do the parameters in Table 1 mean? These need to be explained.
3. Authors should also explain how the rules in the FIS were determined, how many experts’ opinions did they receive, and how many rules are defined for FIS.
4. Are the inputs in the crisp evaluations numeric values or again linguistic expressions? If they are linguistic and the FIS rules are defined according to the same linguistic scale, it is not surprising that the FIS results and the expert opinion results are the same. In my opinion, the similarities of the FIS and the crisp model are a valuable finding only when a numerical scale is used in the crisp model and a linguistic scale is used in the FIS.
5. The representation of references in the text should be consistent and not contain repetitions in the list (For example, Ref 5 and ref 13 are the same, ref 6 and ref 15 are the same)
6. Please make sure that all Figures and Tables in the article are explained.
Author Response
Reviewer#3, Concern # 1:
The fuzzy numbers in column 3 of Table 1 are difficult to understand. It would be easier to read if commas were used.
Author response:
Thank you for your insight. The table is modified as you’ve suggested. We have separated the values by commas.
Author action: We updated the manuscript by Page 7 Table 1.
Reviewer#3, Concern # 2:
If brief information about fuzzy logic was given, the article would be easier to follow, especially for readers unfamiliar with fuzzy systems. What is the membership function, what do the parameters in Table 1 mean? These need to be explained.
Author response:
Thanks for your suggestions. The introduction to fuzzy logic has been briefly explained, and we have also added more information about membership functions and explained the parameters of Table 1.
Author action: We updated the manuscript by Page 7 Section3.
Reviewer#3, Concern # 3:
Authors should also explain how the rules in the FIS were determined, how many experts’ opinions did they receive, and how many rules are defined for FIS.
Author response:
We're grateful for the time you took to review our paper thoroughly. Thank you for your careful reading and attention to detail. For creating FIS rules, we consulted experts, and all the rules were created under the supervision of the experts. We use all the linguistic terms to define the rules and combine them mathematically. We have taken the number of linguistic terms with two values separately and three values separately, and then we defined the rules by this mathematical expression:
32 * 23 = 72
Author action: We updated the manuscript by Page 7 Section3.
Reviewer#3, Concern # 4:
Are the inputs in the crisp evaluation’s numeric values or again linguistic expressions? If they are linguistic and the FIS rules are defined according to the same linguistic scale, it is not surprising that the FIS results and the expert opinion results are the same. In my opinion, the similarities of the FIS and the crisp model are a valuable finding only when a numerical scale is used in the crisp model and a linguistic scale is used in the FIS.
Author response:
Thank you for reading our paper so carefully, we really appreciate it. Some of our inputs can only be answered with a yes or no, such as sweating, where there is only a possibility of whether the person sweats or not. Experts suggested the linguistic scale we have developed. The inputs are taken only in linguistic terms, but the model's procedure is based on the numerical range of the membership functions we have defined.
Author action: We updated the manuscript by Page 7 Section3.
Reviewer#3, Concern # 5:
The representation of references in the text should be consistent and not contain repetitions in the list (For example, Ref 5 and ref 13 are the same, ref 6 and ref 15 are the same.
Author response:
Thank you for your insight. We have cross-checked all the references and removed any duplicates.
Author action: We updated the manuscript by Page # 17 and Page # 18 in the References Section.
Reviewer#3, Concern # 6:
Please make sure that all Figures and Tables in the article are explained.
Author response:
Thank you for your insight. We have properly explained all the tables and figures in the manuscript.
Author action: We updated the manuscript by Page 7 Section 3, Page 9 Section 3, Page 11 Section 4 and Page 14 Section 4.
Reviewer 3 Report
The topic of the paper is interesting, but there are some points that need to be addressed to improve the quality of the paper.
1-The abstract should be expanded to a more clear contribution. It should briefly explain the following: background, motivation, and justification for the research, results, and contribution. The contribution should be clearly presented.
2-The motivation of this paper also should be mentioned.
3-What are the limitations of the proposed model?
Author Response
Reviewer#4, Concern # 1:
The abstract should be expanded to a clearer contribution. It should briefly explain the following: background, motivation, and justification for the research, results, and contribution. The contribution should be clearly presented.
Author response:
I really appreciate your input; thank you for your suggestions. We have updated the abstract according to the requirements specified.
Author action: We updated the manuscript by Page 1, Abstract.
Reviewer#4, Concern # 2:
The motivation of the paper should also be mentioned.
Author response:
Thank you for your insight. The motivation behind our research and the use of fuzzy decision support system has been illustrated in detail.
Author action: We updated the manuscript by Page 2, Section 1.
Reviewer#4, Concern # 3:
What are the limitations of the proposed model.
Author response:
Thanks for your suggestions. The limitations of the proposed system have been explained in the conclusion and future work.
Author action: We updated the manuscript by Page 15, Section 6.
Reviewer 4 Report
- The authors should have included line numbers for the ease of review and the ease of following their responses.
- The writing style need review and improvement. For example, in the abstract, the authors starts with an introduction about the contribution then go back to present awhat is anxiety and why it is important. I believe this should be reversed in order.
- Similar issues arise throughout the paper. In the introduction, it starts with mentioning several psychological factors, then startes that "another factor that affects behavior is anxiety" but anxiety is not another factor, it was mentioned in the list.
- First paragraph of the introduction, several statements need proper referencing (e.g., "Anxiety is a disorder that can
be brought on by ongoing stress, and if it does, people may face significant hazards.", who said so?)
- First paragraph of the introduction, what is UNECE? this is not proper citation style in writing.
- The following sentence, "Numerous studies have shown that anxiety and neuroticism are related", which studies?
- "Disorders like anxiety are highly neglected in Asian countries like Pakistan" need propoer citation and this is the first time either in the abstract that Pakistan is mentioned, the authors need to indicate beforehand (in the abstract and title) that this study relates to Pakistan.
-"After gaining knowledge about the problem, the questions that came to mind were:" this is not scientific writing.
- Grammer error "The main contribution of the paper is highlights".
- "A Fuzzy Inference System is proposed, which will take the symptoms as inputs and predict the outcome based on them". This is not a propoer description.
- Similarly the whole contribution paragraph needs rewriting.
- The authors immediately move to review AI systems, which seems disconnected from the previous introduction.
- Page 3 is mostly copied from other sources.
- This sentence is hard to understand "They searched databases about the diagnostic and statistical manual of mental disorders (DSM-5), one of the categories of anxiety disorders; Generalized Anxiety Disorder, Separation Anxiety Disorder, Social Anxiety Disorder, and Panic Disorder between 2015-2021."
- Figure one is plagarized from the Internet. The authors should have indicated the source or drawn their own figure.
- Figure 3, "mamdani" seems out of place here.
- "MATLAB is a widely used tool in scientific computing, image processing, data visualization, and numerical computation. In the experimental results being discussed,
MATLAB is used to calculate the results. The figures, specifically Figure 8 to Figure 11,
illustrate the use of MATLAB in determining the level of anxiety based on certain factors
such as nervousness, sweating, trembling, increased heart rate, and panic."
This is description is not needed. The discussion of figures 8-11 is very shallow.
- There is no proper description of the data.
- There is no proper evaluation.
- There is no cases that did not display anxiety.
- The table of abbreviations is missing but required by the journal template. Also, the format of the references is wrong.
Author Response
Reviewer#5, Concern # 1:
The authors should have included line numbers for the ease of review and the ease of following their responses.
Author response:
Thank you for your insight. We have added the line numbers in the whole document.
Author action: We updated the manuscript by … N/A
Reviewer#5, Concern # 2:
The writing style need review and improvement. For example, in the abstract, the authors start with an introduction about the contribution then go back to present what is anxiety and why it is important. I believe this should be reversed in order.
Author response:
Thank you for reading our paper so carefully. We really appreciate it. We have carefully updated the whole manuscript and improved the writing style. Furthermore, we have also reversed the order as you suggested.
Author action: We updated the manuscript by Page 1, abstract.
Reviewer#5, Concern # 3:
Similar issues arise throughout the paper. In the introduction, it starts with mentioning several psychological factors, then starts that "another factor that affects behavior is anxiety" but anxiety is not another factor, it was mentioned in the list.
Author response:
We're grateful for the time you took to review our paper thoroughly. Thank you for your careful reading and attention to detail. We have improved the manuscript by excluding anxiety from the list of several psychological factors mentioned initially. Anxiety is now presented as another factor affecting behavior, which we discuss further in the manuscript.
Author action: We updated the manuscript by Page 1 and 2, Section1.
Reviewer#5, Concern # 4:
First paragraph of the introduction, several statements need proper referencing (e.g., "Anxiety is a disorder that can be brought on by ongoing stress, and if it does, people may face significant hazards.", who said so?)
Author response:
Thank you for reading our paper so carefully. We really appreciate it. We have properly added all the missing references in the introduction and literature review section.
Author action: We updated the manuscript by Page 1 and 2, Section 1 and Page 2 and 3, Section 2.
Reviewer#5, Concern # 5:
First paragraph of the introduction, what is UNECE? this is not proper citation style in writing.
Author response:
Thank you for your insight. It was a typing mistake and has been removed from the manuscript. Further, we used Grammarly software to avoid further mistakes.
Author action: We updated the manuscript by Page 1, Section 1.
Reviewer#5, Concern # 6:
The following sentence, "Numerous studies have shown that anxiety and neuroticism are related", which studies?
Author response:
Thank you for your insight. We have added proper references demonstrating the relationship between anxiety and neuroticism.
Author action: We updated the manuscript by Page 1, Section 1.
Reviewer#5, Concern # 7:
Disorders like anxiety are highly neglected in Asian countries like Pakistan" need proper citation and this is the first time either in the abstract that Pakistan is mentioned, the authors need to indicate beforehand (in the abstract and title) that this study relates to Pakistan.
Author response:
I appreciate your valuable perspective and would like to thank you for your insight. We have added proper citations to show that disorders such as anxiety are highly neglected in Asian countries like Pakistan. Furthermore, we have updated the title and abstract.
Author action: We updated the manuscript by Page1 Title, Abstract and Page 1 and 2 Section 1.
Reviewer#5, Concern # 8:
After gaining knowledge about the problem, the questions that came to mind were:" this is not scientific writing.
Author response:
Indeed, your suggestion was very insightful. We have updated the lines to improve readability in the scientific writing style.
Author action: We updated the manuscript by Page2, Section 1.
Reviewer#5, Concern # 9:
Grammar error "The main contribution of the paper is highlights
Author response:
Thank you for your insight. The grammatical error has been corrected and we used Grammarly software to avoid further mistakes.
Author action: We updated the manuscript by … N/A
Reviewer#5, Concern # 10:
A Fuzzy Inference System is proposed, which will take the symptoms as inputs and predict the outcome based on them". This is not a proper description
Author response:
Thank you for your insight. The lines have been carefully updated to enhance their readability.
Author action: We updated the manuscript by Page 2, Section 1.
Reviewer#5, Concern # 11:
Similarly, the whole contribution paragraph needs rewriting.
Author response:
Thank you for reading our paper so carefully, we really appreciate it. We have updated the whole contribution paragraph for better understanding and readability.
Author action: We updated the manuscript by Page 2, Section 1.
Reviewer#5, Concern # 12:
The authors immediately move to review AI systems, which seems disconnected from the previous introduction.
Author response:
We're grateful for the time you took to review our paper thoroughly. Thank you for your careful reading and attention to detail. Our study encompasses two main concepts the medical concept of "anxiety" and our AI system. After discussing anxiety and its causes, we proceed to explain the AI system that we are developed for treatment of anxiety.
Author action: We updated the manuscript by Page 2 and 3 Section 2.
Reviewer#5, Concern # 13:
Page 3 is mostly copied from other sources.
Author response:
Thank you for your insight. We have updated the mentioned section and added appropriate references.
Author action: We updated the manuscript by Page 2 and 3 Section 2.
Reviewer#5, Concern # 14:
This sentence is hard to understand "They searched databases about the diagnostic and statistical manual of mental disorders (DSM-5), one of the categories of anxiety disorders; Generalized Anxiety Disorder, Separation Anxiety Disorder, Social Anxiety Disorder, and Panic Disorder between 2015-2021.
Author response:
Thanks for your suggestions. We have updated the said lines for better understanding and readability.
Author action: We updated the manuscript by Page 2 and 3, Section 2.
Reviewer#5, Concern # 15:
Figure one is plagiarized from the Internet. The authors should have indicated the source or drawn their own figure.
Author response:
Thank you for your insight. We have added the proper citation for Figure One in our research paper.
Author action: We updated the manuscript by Page 4, Section 2.
Reviewer#5, Concern # 16:
MATLAB is a widely used tool in scientific computing, image processing, data visualization, and numerical computation. In the experimental results being discussed. MATLAB is used to calculate the results. The figures, specifically Figure 8 to Figure 11, illustrate the use of MATLAB in determining the level of anxiety based on certain factors such as nervousness, sweating, trembling, increased heart rate, and panic.
Author response:
I really appreciate your input; thank you for your suggestions. We have added descriptions for the result in Figures 8 to 11 to improve the readability and understanding of the readers.
Author action: We updated the manuscript by Page 11, section 4.
Reviewer#5, Concern # 17:
The discussion of figures 8-11 is very shallow. There is no proper description of the data. There is no proper evaluation. There is no cases that did not display anxiety. The table of abbreviations is missing but required by the journal template. Also, the format of the references is wrong.
Author response:
Thank you for reading our paper so carefully. We really appreciate it. We have added proper descriptions of the data and improved the explanation of the evaluation part for better understanding and readability. We have also included cases that show no anxiety, added appropriate abbreviations, and updated the reference format according to the journal guidelines.
Author action: We updated the manuscript by Page 11, section 4, and Page 14, section 4, Page # 17, and Page # 18 in the References Section.
Round 2
Reviewer 1 Report
It can be accepted now
Author Response
Thank you for your feedback. Once again, we sincerely appreciate your constructive feedback and guidance throughout the review process. Your input has been instrumental in improving the quality and clarity of our work.
Reviewer 2 Report
Thank you for the corrections.
Author Response

(The authors gave the same response as above.)

Reviewer 4 Report
The authors ignored my most pressing comments regarding:
- There is no proper description of the data.
- There is no proper evaluation.
- There is no cases that did not display anxiety.
Author Response
Reviewer#4, Concern # 1:
There is no proper description of the data.
Author response:
Thank you for your feedback regarding the description of the data in our manuscript. We apologize for any confusion or oversight in our initial submission. We have taken your comment into serious consideration and have made the necessary revisions to address this concern.
Author action: We updated the manuscript by Page 14, Section 4.
Reviewer#4, Concern # 2:
There is no proper evaluation.
Author response:
We appreciate your feedback regarding the evaluation section of our manuscript. Thank you for highlighting this concern. We have carefully reviewed your comment and made the necessary revisions to address it.
Author action: We updated the manuscript by Page 14, 15 Section 4.
Reviewer#4, Concern # 3:
There is no proper evaluation.
Author response:
We greatly appreciate your feedback regarding the absence of cases that did not display anxiety in our manuscript. Thank you for bringing this concern to our attention. We have carefully considered your comment and have made the necessary additions to address this issue.
Author action: We updated the manuscript by Page 14, 15 Section 4.